# Chemical Composition, Antitumor Potential, and Impact on Redox Homeostasis of the Essential Oils of *Orlaya grandiflora* from Two Climate Localities

**DOI:** 10.3390/molecules27185908

**Published:** 2022-09-11

**Authors:** Milena D. Vukic, Ana D. Obradovic, Nenad L. Vukovic, Miroslava Kačániová, Predrag M. Djurdjevic, Gorica T. Djelic, Milos M. Matic

**Affiliations:** 1Department of Chemistry, Faculty of Science, University of Kragujevac, Radoja Domanovića 12, 34000 Kragujevac, Serbia; 2Department of Biology and Ecology, Faculty of Science, University of Kragujevac, Radoja Domanovića 12, 34000 Kragujevac, Serbia; 3Institute of Horticulture, Faculty of Horticulture and Landscape Engineering, Slovak University of Agriculture in Nitra, Tr. A. Hlinku 2, 94976 Nitra, Slovakia; 4Department of Bioenergy, Food Technology and Microbiology, Institute of Food Technology and Nutrition, University of Rzeszow, 4 Zelwerowicza St, 35601 Rzeszow, Poland; 5Department of Internal Medicine, Clinic for Hematology Clinical Center Kragujevac, Faculty of Medical Science, University of Kragujevac, 34000 Kragujevac, Serbia

**Keywords:** *Orlaya grandiflora*, essential oils, chemical composition, cancer cell lines, antitumor potential, redox homeostasis, cell migration

## Abstract

It is well known that abiotic components can affect biosynthetic pathways in the production of certain volatile compounds. The aim of this study was to compare the chemical composition of essential oils obtained from *Orlaya grandiflora* (L.) Hoffm. collected from two localities in Serbia (continental climate, OG1) and Montenegro (Mediterranean climate, OG2) and to assess their antitumor potential on the human colon cancer HCT-116 and breast cancer MDA-MB-231 cell lines. EOs obtained by hydrodistillation were analyzed using GC-MS and GC-FID methods. The results indicate considerable differences in the chemical compositions of the two samples. Although in both samples the main class of volatiles observed was sesquiterpenes (47.5% for OG1 and 70.1% for OG2), the OG1 sample was characterized by a high amount of monoterpene hydrocarbons (29.3%), and sesquiterpene germacrene D (29.5%) as the most abundant compound. On the other hand, the OG2 sample contained a high quantity of oxygenated sesquiterpenes (20.6%), and β-elemene (22.7%) was the major constituent. The possible antitumor mechanisms of these EOs in the HCT-116 and MDA-MB-231 cell lines were examined by means of cell viability, apoptosis, redox potential, and migratory capacity. The antiviability potential appeared to be dose dependent, since the results showed that both EOs decreased the viability of the tested cells. Stronger antitumor effects were shown in MDA-MB-231 cells after short-term treatment, especially at the highest applied concentration, where the percentage of viability was reduced by over 40%. All tested concentrations of EOs exhibited proapoptotic activity and elevated activity of caspase-3 in a dose- and time-dependent manner. The results also showed decreased concentrations of superoxide anion radical in the treated cells, which indicates their significant antioxidative role. Long-term treatments showed mild recovery effects on cell viability in both cell lines, probably caused by the balancing of redox homeostasis. Elevated levels of nitrites indicate high levels of nitric oxide (NO) production and suggest its higher bioavailability due to the antioxidative environment. The tested EOs also induced a drop in migratory capacity, especially after short-time treatments. Taken together, these results suggest considerable antitumor activity of both EOs, which could have potential therapeutic applications.

## 1. Introduction

Essential oils represent a liquid extract obtained from aromatic plants that consists of a complex mixture of volatile, lipophilic plant secondary metabolites. These secondary metabolites have a strong odor responsible for plants’ specific scent. Chemically, they belong to the classes of terpenes, terpenoids, phenolics, fatty acids degradation products, amino acid-derived products, phenylpropanoids, and hydrocarbons formed by diverse biogenetic pathways (synthesized in plant organs, particularly flowers, buds, leaves, seeds, stems, and fruits) [1,2,3,4,5,6]. Their crucial role in plant life involves attracting valuable insects and other pollinators and protecting plants from herbivores, pathogens, and environmental stress. In the industry, essential oils are commonly used in cosmetics, agriculture, sanitary products, and aromatherapy, since many of them are generally recognized as safe (GRAS) by both the US Food and Drug Administration (FDA, White Oak, MD, USA) and the Environmental Protection Agency (EPA, Washington, DC, USA) [1,2,4]. Essential oils are also known for their confirmed wide range of biological activities, such as antimicrobial, against a large number of bacteria and fungi; antitumor and proapoptotic; antioxidant; antiviral; antimycotic; antiparasitic; and insecticidal, depending on their chemical composition and environmental and plant genotype factors [1,2,5,7].

Cancer represents one of the most persistent groups of diseases with high mortality worldwide. Despite the extensive use of multimodal chemotherapies, their efficiency is limited, emphasizing the need for novel therapeutic approaches with higher cytotoxicity against malignant cells and acceptable harmful outcomes in healthy tissues. Naturally occurring compounds such as terpenoids, phenolics, flavonoids, and alkaloids exert significant therapeutic effects [8].

Reactive oxygen and nitrogen species (ROS and RNS) may induce carcinogenesis and progression of the tumor by genetic and epigenetic pathways, and the presence of oxidative stress has been detected in various tumors [9].

The Apiaceae, parsley, or carrot family includes 3780 species and 434 genera, mainly distributed in the Mediterranean and Southwest Asia, and is considered one of the oldest families among aromatic flowering plants [5,10,11,12,13]. Species belonging to this family have been used from ancient times by humans for food (vegetables, herbs, and spices), as well as for medicinal purposes [14,15,16]. Apiaceae species are characterized by an aromatic smell due to the presence of essential oil or oleoresin, so they also have application potential in aromatherapy [16].

*Orlaya grandiflora* (L.) Hoffm. white lace flower grows in Europe (southern, central, and western Europe), in the Caucasus region, and in central and southwestern Asia, in a dry and warm habitat [11]. So far, there has been no extensive research about this plant’s chemical composition and biological activity. Ethnobotanical studies indicate a laxative effect of the aboveground part of the plant prepared in the form of a decoction [17]. Based on Pavel’s research (2011), it is indicated that *O. grandiflora* extract has an insecticidal and inhibitory effect on the growth of *Spodoptera littoralis* larvae [18]. Previous reports on *O. grandiflora* essential oils have indicated that abundant compounds are sabinene, α-pinene, γ-terpinene, β-caryophyllene, and germacrene D, while in the essential oils of the fruit of this plant as majors were identified sesquiterpenes β-caryophyllene, δ-cadinene, α-amorphene, and germacrene D [5,10].

It is well known that plants of the same species that grow in diverse environments may have alterations in the concentration of secondary metabolites. Abiotic factors, including water, light, temperature, soil, and chemicals, have a great influence on the production and accumulation of volatiles [19]. In light of the above mentioned, the aim of this study was to examine and compare the chemical composition of *O. grandiflora* essential oils collected from Serbia (continental climate) and from Montenegro (Mediterranean climate). For this purpose, we obtained essential oils from the aerial parts of plants using Clevenger-type apparatus, and obtained oils were characterized using GC-MS and GC-FID analysis. In addition, we examined the possible antitumor mechanisms, including biocompatibility (on MRC-5 cell line), cell viability, proapoptotic effects, impact redox homeostasis, and migratory capacity of the obtained essential oil on the human colon cancer HCT-116 and breast cancer MDA-MB-231 cell lines.

## 2. Results and Discussion

### 2.1. Chemical Composition of OG1 and OG2

After performing hydrodistillation, the contents of essential oil in OG1 (Ovčar-Kablar Gorge; continental climate) and OG2 (Budva, Jaz, Mediterranean climate) were 0.07% (*v*/*w*) and 0.09% (*v*/*w*), respectively. Chemical compositions of both essential oil samples are presented in Table 1, amounts of hydrocarbon and oxygenated terpenes in Table 2, while Table 3 shows the class and percentage amounts of each individual subclass of terpenes. In the oils from both plant samples (Table 1), 86 volatile compounds were identified; among them, 67 constituents (98.2%) in OG1 and 73 constituents (98.7%) in OG2 were detected, respectively. The main constituents of essential oil OG1 were germacrene D (29.5%), sabinene (11.5%), terpinen-4-ol (9.6%), (*E*)-caryophyllene (5.5%), α-pinene (4.5%), β-pinene (4.1%), and γ-terpinene (3.6%). Other volatiles were observed in quantities less than 2.6%. Quite different results were observed for the plant sample collected from the Mediterranean region of Montenegro (OG2). The most abundant compounds were β-elemene (22.7%; only 1.1% was observed from plant material collected from the continental part of Serbia), germacrene D (14.3%), germacrene A (12.5%), and (*E*)-caryophyllene (6.8%), while other constituents were identified in amounts less than 2.9%. One previous research related to the chemical analysis of essential oils of the Apioideae taxa revealed that *O. grandiflora* essential oil contains sabinene (25.4%) and α-pinene (9.6%) as major compounds [10].

The results presented in Table 2 clearly indicate that sesquiterpene hydrocarbons represent the most abundant class of terpenes (47.5% and 70.1%, respectively). In addition, essential oil from OG2 with 29 compounds contains a high quantity of oxygenated sesquiterpenes (20.6%). Contrary, in OG1, we found a high amount of monoterpene hydrocarbons (29.3%).

As can be seen from Table 3, the main class of volatiles observed in both oil samples are sesquiterpenes (OG1, 55.5% of the total; OG2, 90.7% of the total). Interestingly, both oil samples contain as the most abundant compounds monocyclic sesquiterpenes (32.7% and 52.8%, respectively), with the fact that OG1 contains a high quantity of constituents with a monocyclic germacrane skeleton (30.9%). On the other hand, OG2 is rich with volatiles with a monocyclic germacrane skeleton (28.3%) and a monocyclic elemane skeleton (22.7%). In addition, it must be noted that both essential oil samples contain a high concentration of bicyclic sesquiterpenes (17.9% and 27.3%, respectively). Contrastively, constituents from the subclass of tricyclic, acyclic, and tetracyclic sesquiterpenes were not quantified in high concentrations (less than 3.1%). From Table 3, it can be observed that essential oil from OG1 (plant material from the continental part of Serbia) has a significant concentration of monoterpenes, especially bicyclic monoterpenes (20.8%). Among them, we should emphasize bicyclic monoterpenes (20.8%) with a bicyclic cyclopropane thujane skeleton, as well as monocyclic monoterpenes (19.1%) with a monocyclic *p*-menthane skeleton. On the other hand, chemical analysis of essential oil obtained from plant material from the Mediterranean region of Montenegro (OG2) shows a low quantity of monoterpenes. In addition, this sample of essential oil contains a low amount of tricyclic and acyclic monoterpenes.

Differences in the chemical compositions of *O. grandiflora* essential oils are caused by the different ecological environments of the collected plant materials. These ecological differences induce different biosynthetic pathways in the production of certain volatiles [20,21,22,23].

The above-mentioned results clearly indicate that essential oils of plant *O. grandiflora* are rich sources of biologically active secondary metabolites, and further biological investigations are necessary to define possible pharmaceutical and therapeutic properties.

### 2.2. Cell Viability Assays

Since there are no previous reports in the literature considering the antitumor activity of any *O. grandiflora* essential oils, our next step was to investigate the effects on the viability of OG1 and OG2. For that purpose, we conducted an MTT cell viability assay on the human fibroblast cell line (MRC-5), to determine the biocompatibility of these essential oils, and on two human tumor cell lines, HCT-116 (colon carcinoma) and MDA-MB-231 (breast adenocarcinoma).

#### 2.2.1. Determination of Biocompatibility

The viability of the treated human fibroblast cell line MRC-5 cells after 24 and 72 h by both essential oils of *O. grandiflora* (OG1 and OG2) was higher than 84%, as presented in Figure 1. The demonstrated results indicate that biocompatibility is acceptable and that the tested samples are not harmful to healthy human fibroblast cells, qualifying this essential oil as suitable for the further biological evaluation of its potential antitumor/antiviability applications.

#### 2.2.2. Determination of Cell Viability (MTT Assay)

With the aim to determine the antiviability effects of the extracted essential oils (OG1 and OG2), an MTT cell viability assay was conducted on the colon cancer (HCT-116) and human breast cancer (MDA-MB-231) cell lines. Results obtained after 24 h and 72 h of incubation with various concentrations of *O. grandiflora* essential oils obtained from the two regions are presented in Figure 2. Generally, the results indicate significant antiviability effects compared to the nontreated cells, and a dose-dependent decrease in cell viability was observed for both cell lines after treatment with OG1 and OG2. The antiviability activity of OG1 and OG2 was the strongest in the highest applied concentration of 200 µg/mL with no significant difference in higher concentrations applied (400 µg/mL and 600 µg/mL), after both incubation times, with the strongest effect demonstrated after short-term (24 h) treatment on both tested cell lines. By comparing the obtained results, we conclude that MDA-MB-231 cells were more sensitive to treatments, especially with OG1 after 24 h of incubation in the highest applied concentration (55%).

The observed antiviability outcomes could be due to the reduced proliferative potential of the tested cells and to the proapoptotic effects of the main constituents of the essential oils. Literature data show that β-elemene inhibits the growth of many tested cancer cells and shows antitumor effects, in vitro and in vivo, by different mechanisms [24,25,26]. For more than 20 years, β-elemene has been applied as an oral emulsion and injection in clinical treatments as an antitumor drug for some cancers [26]. In addition, there are previous reports on germacrene D and sabinene being potent inhibitors of the growth of some human tumor cell lines [27,28]. Based on an investigation by Shapira et al. terpinene-4-ol induced the inhibition of growth of colorectal, pancreatic, prostate, and gastric cancer cells, while *E*-caryophyllene was reported to have significant anticancer activities, affecting the growth and proliferation of numerous cancer cells [29,30]. However, in the case of complex mixtures, it is more likely that the occurring synergy between various components is responsible for the expressed effects.

### 2.3. Apoptotic Effects of O. grandiflora Essential Oils in Tumor Cells

Apoptosis is a highly regulated process of cell death and is crucial in maintaining tissue homeostasis, regulating the cell division ratio, and preventing carcinogenesis. Apoptosis-inducing agents are expected to be successful antitumor drugs since apoptosis is a protective mechanism against cancer development that acts to remove genetically damaged cells from the tissue before they undergo clonal expansion [31]. The components of essential oils from various plants have been indicated to induce apoptosis in numerous cancer cell types. Since our results suggest decreased viability of the tested essential oils, we performed the measurements of the apoptotic potential as one of the mechanisms of recorded antitumor activity in this study. One of the major indicators of proapoptotic potential is caspase-3, a specific protease that leads the execution stage of the programmed cell death process [32,33].

#### 2.3.1. Determination of Cell Apoptosis by Annexin V-FITC/7-AAD Staining

The type of cell death was determined by flow cytometric analysis of the treated cells stained with Annexin V-FITC and 7-AAD. Both cell lines (HCT-116 and MDA-MB-231) were treated for 24 h and 72 h, with concentrations of 10 µg/mL and 100 µg/mL. These two concentrations were chosen as representative of the average low and average high doses applied in the majority of studies, since there were no significant differences in MTT assay results between the doses higher than 100 µg/mL in our study. The effects of OG1 and OG2 essential oils showed a statistically significant time- and dose-dependent proapoptotic effect in HCT-116 and MDA-MB-231 cells, as shown in Figure 3 and Figure 4. The strongest apoptosis level compared to nontreated cells was recorded for OG2 essential oil at the concentration of 100 µg/mL.

#### 2.3.2. Determination of Caspase-3 Activation

Since our results indicate that the essential oils exert antiapoptotic effects on the tested cancer cells, consequently, we evaluated the expression of the main proapoptotic marker, caspase-3, by flow cytometric analysis of the protein expression level of this key regulatory apoptotic protein [34]. The results indicated that both essential oils at both examined concentrations stimulated the activation of caspases-3 in HCT-116 and MDA-MB-231 cells compared to control, as shown in Figure 5. These results are in congruence with the obtained changes in apoptosis levels in the treatments. The strongest rise of caspase-3 activity (3.56-fold) compared to control was detected for OG2 oil at the concentration of 100 µg/mL, which correlates with the highest proapoptotic ratio detected in this study.

### 2.4. Effects of O. grandiflora Essential Oils on Redox Status in Tumor Cells

The results presented so far indicate that essential oils obtained from *O. grandiflora* decrease the viability of the tumor HCT-116 and MDA-MB-231 cell lines and have no effect on the viability of human fibroblast cells (MRC-5). In order to propose the potential mechanism of their effects on cancer cell viability, we also examined the effects of OG1 and OG2 on oxidative stress markers, precisely on the production of superoxide anion radical (O_2_^•−^) and nitrites (NO_2_^−^).

#### 2.4.1. Determination of Superoxide Anion Radical (NBT Assay)

ROS appear to be involved in the regulation of various physiological pathways, including signal transduction, apoptosis, and differentiation. Recently, emerging evidence has suggested the involvement of ROS and the aberrant activation of redox-sensitive signaling pathways in tumor invasion and migration. Some ROS-regulated proteins play key roles in epithelial–mesenchymal transition and tumor metastasis, including the effects on E-cadherins, integrins, and matrix metalloproteinases [35]. Accordingly, in this study, we estimated levels of superoxide anion radical (O_2_^•−^) production by HCT-116 and MDA-MB-231 cells after 24 h and 72 h of incubation with various concentrations of OG1 and OG2, and the obtained results are presented in Figure 6.

Compared to the control cells, all applied concentrations exhibited a significant reduction of O_2_^•−^ levels in the tested tumor cells, at both time treatments. In HCT-116 cells, the antioxidative potential was the strongest at the lowest applied concentrations (22% and 27% reduction compared to control/time, respectively). On the contrary, in MDA-MB-231 cells, the strongest antioxidative effect of oils was recorded at the highest applied concentration of OG2 (30% and 20% reduction compared to control/time, respectively). Some antioxidants may enhance the effects of cytotoxic regimes, improving the response rate of the tumor to chemotherapeutic agents, while some others can ameliorate their antitumor activity [36,37]. Since one of the major hallmarks of cancer cells is the presence of oxidative stress and disbalance of redox homeostasis, our results indicate that the exerted antioxidant impact could be an important pathway in the regulation of cancer cell progression and viability. The precise antitumor outcome depends on tumor type and applied dose, suggesting the need for personalized usage.

#### 2.4.2. Determination of Nitrites (Griess Assay)

Nitric oxide (NO) is an important signaling molecule in numerous physiological and pathological conditions. NO is reported to have antitumor activities, as well as protumor properties, depending on the timing, concentration, and tissue type [38]. Therefore, we evaluated the production of nitrites in the HCT-116 and MDA-MB-231 cell lines after 24 h and 72 h of incubation with OG1 and OG2, and the results are presented in Figure 7.

Treatment with both essential oils showed a significant increase in the production of nitrite by HCT-116 cells compared to the control. Low concentrations of NO can stimulate cell growth and protect many cell types from apoptosis, whereas its high concentrations can inhibit cell growth and induce apoptosis [39]. The changes in the production of NO could affect various signaling pathways that involve nitric oxide, leading to potential antitumor outcomes. Superoxide anion radical reacts with NO at high affinity, forming aggressive peroxynitrite anion (ONOO^−^), leading to nitrosative stress in cells. Since NO has a half-life of only several seconds in a solution rich in superoxide anion radical [40], a prooxidative environment depletes NO bioavailability, while in a surrounding with a low level of superoxide, anion radical NO has much greater stability and prolonged signaling effects [41]. In HCT-116 cells, all applied concentrations, at both time treatments, induced an increase in NO production compared to control, especially in the lowest concentration of OG2 (34% increase), which correlates with the strongest antioxidative effects determined by O_2_^•−^ production in these cells. The obtained data indicate that stimulation in NO production and/or bioavailability significantly contributes to the recorded antitumor activity of the tested essential oils.

In MDA-MB-231 cells, the long-term treatment decreased the production of NO at all concentrations, which corresponds to the lower detected reduction in cell viability after long-term treatment compared to short-term treatment in this cell line.

Generally, the investigated essential oils exerted a considerable decrease in viability of the tested cancer cell lines and showed significant antioxidative potential correlated with decreasing O_2_^•−^ levels. In addition, the majority of the concentrations suggested elevated production of nitric oxide and significant dose-dependent cytotoxic effects. The obtained data indicate that the tested compounds are suitable for further investigations in designing novel antitumor therapy.

### 2.5. Transwell Assay for Cell Migration

To examine the effects of OG1 and OG2 essential oils treatments on the migration capacity of cancer cells, a 2D transwell migration assay was performed. The results indicate a significant dose-dependent decrease in the cell migration index of both HCT-116 and MDA-MB-231 cells exposed to OG1 and OG2 oils compared to the nontreated cells, as presented in Figure 8. Short-term exposure (24 h) of 100 µg/mL OG1 essential oil to HCT-116 cells exerted the strongest decrease in migration capacity of 40% compared to the control. Based on our results regarding ROS production, we can suggest that antioxidative potential could elevate nitric oxide bioavailability leading to antimigratory outcomes recorded in our study, since numerous studies indicate that nitric oxide can inhibit cell migration [42].

## 3. Materials and Methods

### 3.1. Reagents and Chemicals

In this study, the following reagents and chemicals were used: Dulbecco’s Modified Eagle Medium (DMEM), 10% fetal bovine serum (FBS), 0.4% Trypan blue, 0.25%, trypsin-EDTA, dimethyl sulfoxide (DMSO), 3-(4,5-Dimethylthiazol-2-yl)-2,5-diphenyltetrazolium bromide (MTT), phosphate-buffered saline (PBS), nitroblue tetrazolium chloride (NBT), 0.1% N-(1-naphthyl)ethylenediamine dihydrochloride (NED), 0.1 M sodium nitrite (NaNO_2_), 1% sulfanilamide (SAA) dissolved in phosphoric acid (H_3_PO_4_), 4% paraformaldehyde, 0.1% crystal violet in 200 mM 2-(N-Morpholino) ethanesulfonic acid, and 10% acetic acid. All the chemicals and reagents used in this study were of the highest commercially available purity.

### 3.2. Collection, Identification, and Preparation of Plant Samples

The aerial parts of *O. grandiflora* were collected from Ovčar-Kablar Gorge (GPS 43.907389 N 20.202534 E, Serbia, June 2018, Voucher No. 50/17) OG1 and Budva, Jaz (GPS 42.285324 N 18.78334 E, Montenegro, June 2018, Voucher No. 51/17) OG2. Botanical identification and depositions of examined plant materials were performed in the Herbarium of the Department of Botany, Faculty of Biology, University of Belgrade, Serbia (17130, BEOU).

Prior to the isolation of essential oils, both plant samples of *O. grandiflora* were dried at room temperature in shade for three weeks. Finally, both plant materials were ground by using a grinder.

### 3.3. Isolation of Essential Oil Samples

The essential oils from two samples of *O. grandiflora* were obtained by hydrodistillation in a Clevenger-type apparatus for 3 h. The obtained oils were dried over anhydrous sodium sulfate. Until GC-MS and GC-FID examinations, oil samples were stored in sealed vials in the dark at −4 °C.

### 3.4. Identification of Volatile Constituents by Gas Chromatography (GC) and Gas Chromatography/Mass Spectrometry (GC/MS)

Gas chromatography (GC) and gas chromatography–mass spectrometry (GC-MS) analyses were performed using an Agilent 7890A GC equipped with inert 5975C XL EI/CI MSD and an FID detector connected by a capillary flow technology 2-way splitter with make-up (to MSD: capillary column 1.44 m × 180 μm × 0 μm at 325 °C; to FID: capillary column 0.53 m × 180 μm × 0 μm at 325 °C). An HP-5MS capillary column (30 m × 0.25 mm × 0.25 μm) was used. The temperature of the GC oven was programmed from 60 °C to 270 °C with an increasing rate of 3 °C/min. Finally, the temperature was increased to 300 °C (for 30 °C/min), with a holding time of 4 min. Helium 5.0 was used as the carrier gas with a flow rate of 1 mL/min. The injection volume was 1 μL, while the split/splitless injector temperature was set at 250 °C. With a split ratio of 10:1, the investigated samples were analyzed in the split mode. The FID or GC detector temperature was set at 300 °C. Electron-impact mass spectrometric data (EI-MS; 70 eV) were acquired in scan mode over the *m*/*z* range of 40–550. MSD transfer line, ion source, and quadrupole temperatures were 315 °C, 230 °C, and 150 °C, respectively. Acquisition of data started after a solvent delay time of 3 min. The components were identified based on their retention indices and comparison with reference spectra (Wiley and NIST databases) [43]. The retention indices were experimentally determined using the standard method, which included retention times of n-alkanes (C6–C34), injected under the same chromatographic conditions [44]. The percentages of the identified compounds (amounts higher than 0.1%) were derived from their GC peak areas.

### 3.5. Cell Culture and Treatment

The human lung fibroblast cell line MRC-5, colon cancer cell line HCT-116, and breast cancer cell line MDA-MB-231 were obtained from the American Tissue Culture Collection. These cells were propagated and maintained in DMEM and supplemented with 10% FBS and a combination of antibiotics (100 IU/mL penicillin and 100 µg/mL streptomycin). The cells were seeded in a 96-well microplate (10,000 cells per well) and cultured in a humidified atmosphere with 5% CO_2_ at 37 °C. Two essential oils of *O. grandiflora* were used in experiments. After 24 h of cell incubation, 100 μL of medium containing various doses of treatment (1 µg/mL, 10 µg/mL, 20 µg/mL, 50 µg/mL, 100 µg/mL, 200 µg/mL, 400 µg/mL and 600 µg/mL) was added to each well of the microplate, and the cells were incubated for 24 h and 72 h, after which the evaluation of cell viability was performed. The parameters of redox status, superoxide anion radical, and nitrites were evaluated after exposure to six increasing doses of essential oils (1 µg/mL, 10 µg/mL, 20 µg/mL, 50 µg/mL, 100 µg/mL and 200 µg/mL). The estimation of the effects of the tested oils on proapoptotic activity, activity of caspase-3, and migratory potential was performed after the treatment with two selected concentrations (10 µg/mL and 100 µg/mL). Non-treated cells were used as control. The stock solutions were prepared from the concentration of 10 mg/mL made in a DMEM/DMSO mixture in a ratio of 9:1 (*v*/*v*). All treatment concentrations were obtained by serial dilutions of stock solution, so DMSO concentrations decreased continuously. All treatments were performed in triplicate for each assay.

### 3.6. Determination of Cell Viability (MTT Assay)

The cell viability was determined by using an MTT assay [45]. The cells were plated at a density of 100,000 cells/mL (100 µL/well) in 96-well microplates in DMEM. After incubation (24 h), the 8 different concentrations of both essential oils (1 µg/mL to 600 µg/mL) were added to each well (100 µL/well). The untreated cells (cultured only in DMEM) served as control. After 24 and 72 h of incubation, the cell viability was determined by MTT assay. A 20 µL volume of MTT (concentration of 5 mg/mL) was added to each well, and after 3 h of incubation, the formed formazan crystals were dissolved by adding 20 µL of DMSO. The obtained color was measured on an ELISA reader at the wavelength of 550 nm. The percentage of viable cells was calculated as the ratio between the absorbance at each dose of the treatment and the absorbance of the control multiplied by 100 to obtain a percentage.

### 3.7. Determination of Type of Cell Death

Apoptosis and necrosis were analyzed by double staining with Annexin V-FITC and 7-AAD. Annexin V binds to the cells with exposed phosphatidylserine, whereas 7-AAD labels the cells with membrane damage. Apoptotic cells were detected using the Annexin V-FITC/7-AAD Kit (Apoptosis Detection Kit, Beckman Coulter, Brea, CA, USA). Staining was performed according to the manufacturer’s instructions and Shounan protocol [46]. After the treatment with OG1 and OG2 at a concentration of 10 µg/mL and 100 µg/mL, the cells were collected, washed in PBS, and resuspended in ice-cold binding buffer. Ten thousand events were analyzed on Flow Cytometer Cytomics FC500 (Beckman Coulter, USA). The percent of viable (Annexin V^−^7-AAD^−^) cells, early apoptotic (Annexin V^+^7-AAD^−^) cells, late apoptotic (Annexin V^+^7-AAD^+^) cells, and necrotic cells (Annexin V^−^7-AAD^+^) cells were evaluated by Flowing Software (http://www.flowingsoftware.com/, accessed on 27 January 2021).

### 3.8. Determination of Caspase-3 Activation

For flow cytometric analysis experiments, HCT-116 and MDA-MB-231 cells were plated in 6-well microtiter plates at a density of 500,000 per well and incubated overnight for cell attachment and recovery. Subsequently, the cells were exposed to treatments OG1 and OG2 in concentrations of 10 µg/mL and 100 µg/mL values or DMEM supplemented (control) for 24 h and 72 h at 37 °C in an atmosphere of 5% CO_2_ and absolute humidity. After treatment cells were collected, washed in PBS, fixed, and permeabilized using the Fixation and Permeabilization Kit (eBioscience, San Diego, CA, USA). In order to verify apoptosis, the cells were incubated with a primary anti-cleaved caspase-3 antibody (Cell Signaling Technology, Danvers, MA, USA) for 20 min at room temperature. Then, the cells were washed in PBS and stained with a secondary FITC-conjugated antibody (Abcam, USA). The samples were analyzed by Flow Cytometer Cytomics FC500 (Beckman Coulter, USA), and the expression of active caspase-3 was evaluated by Flowing Software (http://www.flowingsoftware.com/, accessed on 27 January 2021).

### 3.9. Measurement of Superoxide Anion Radical (NBT Test)

The concentration of superoxide anion radical (O_2_^•−^) in samples is based on the reduction of nitroblue tetrazolium (NBT) to nitroblue-formazan in the presence of O_2_^•−^ [47]. The assay was performed by adding 20 μL of 5 mg/mL NBT to each well, followed by cell incubation for 1 h, after which the formazan was solubilized by adding 20 μL of DMSO. The absorbances were measured on an ELISA reader at 550 nm. The concentrations of O_2_^•−^ were expressed as nmol O_2_^•−^/mL in 10^5^ cells/mL.

### 3.10. Measurement of NO Concentration (Griess Method)

The spectrophotometric determination of nitrites (NO_2_^−^) as an indicator of the nitric oxide (NO) level was performed by using the Griess method [48]. The concentration of NO_2_^−^ is directly proportional to the intensity of the purple color measured on the reader. The Griess reaction is based on the coupling of NO-generated diazonium ion with N-(1-naphthyl) ethylenediamine where a chromophoric product is formed. Equal volumes of 0.1% (1 mg/mL)N-1-naphthyl ethylenediamine dihydrochloride and 1% (10 mg/mL) sulfanilamide solution in 5% phosphoric acid were mixed to form the Griess reagent just before the application to the microplate. During the 10 min of incubation (at room temperature, protected from the light sources) purple color was developed. After incubation, absorbances were measured on an ELISA reader at 550 nm, and the nitrite concentration was expressed in μmol NO_2_^−^/mL in 10^5^ cells/mL.

### 3.11. Transwell Assay for Cell Migration

The cell migration capacity was determined by the ability of cells to pass the pores of polycarbonate membranes (pore size 8 µm; Greiner Bio-One, Gallen, Switzerland) at the bottom of transwell chambers. The migration test was performed according to the protocol described by Chen [49]. The cells were exposed to 10 µg/mL and 100 µg/mL concentrations of treatments OG1 and OG2 for 24 h and 72 h, respectively. The control cells were cultured only in DMEM. After the treatment exposures, all groups of treated cells were trypsinized and placed in the upper chambers at a density of 100,000 cells/well in 500 µL of DMEM with 10% FBS. The lower chambers of the control cells contained 750 µL of DMEM supplemented with 10% FBS, whereas the lower chambers with treated cells were filled with 10 µg/mL and 100 µg/mL concentration of treatments OG1 and OG2. After 6 h of incubation at 37 °C, the cells from the upper surface of the filter were completely removed with gentle swabbing. The remaining migrated cells were fixed for 20 min at room temperature in 4% paraformaldehyde and stained with 0.1% crystal violet in 200 mM 2-(N-Morpholino) ethanesulfonic acid (pH 6.0) for 10 min. An amount of 10% acetic acid dissolved the dye, and the absorbance was measured at 595 nm. The migration index was calculated as the ratio of absorbance of the treated samples divided by the absorbance of the nontreated control cell value and multiplied by 100 to give the percentage.

### 3.12. Statistical Analyses

All data were evaluated using IBM-SPSS 23 software for Windows (SPSS Inc., Chicago, IL, USA). The data were presented as a mean ± standard error (S.E.M). The statistical significance was determined using a paired-sample *t*-test. The level of statistical significance was set at * *p* < 0.05.

## 4. Conclusions

In this research, we presented the chemical composition of essential oils of *O. grandiflora* collected from Serbia (characterized by the continental climate, OG1) and Montenegro (characterized by the Mediterranean climate, OG2). Quite large differences in the chemical composition of the examined plant materials were observed. Sesquiterpene germacrene D with 29.5% represents the most abundant constituent of essential oil of OG1, while β-elemene was observed as a major compound found in the plant sample from the Mediterranean region of Montenegro (OG2). Although both samples were characterized by high amounts of sesquiterpene hydrocarbons, OG2 contains a high quantity of oxygenated sesquiterpenes (20.6%). On the other hand, in collected plant material from Serbia (OG1), we observed a high amount of monoterpene hydrocarbons (29.3%). Finally, we confirm that climate conditions (also type of soil) have a big influence on the biosynthesis of individual terpenes. Even though both essential oil samples were characterized by high amounts of monocyclic sesquiterpenes (32.7% and 52.8%, respectively), OG2 contains high concentrations of constituents with monocyclic germacrane and elemane skeletons (30.9% and 22.7%, respectively). Contrastively, in plant material from the continental part of Serbia, we observed high concentrations of compounds from the subclass of a bicyclic cyclopropane thujane skeleton, as well as monocyclic monoterpenes with a monocyclic p-menthane skeleton. According to the chemical composition of the investigated essential oils, they are expected to reduce cancer cell viability, which was demonstrated in our study. Both essential oils decreased the viability of HCT-116 and MDA-MB-231 while not significantly affecting the viability of the human lung fibroblast cell line MRC-5, indicating their favorable biocompatibility. OG1 and OG2 essential oils induced a significant proapoptotic effect on both tested cancer cell lines and increased the activity of caspase-3, as a major proapoptotic marker, suggesting their considerable antitumor potential in these cells. In addition, the tested oils exerted a considerable antioxidative potential in both tested human cancer cell lines, making them a suitable candidate for various cotreatments with existing chemotherapeutics. The tested essential oils exerted antimigratory effects on both tested cancer cell types, which indicates their considerable potential against tumor progression and metastasis. The obtained data could also serve as a basis for the mechanism of action of these essential oils in the further designing of novel antitumor strategies. Since the complete molecular basis of their biological activity is still to be determined, further examinations of the signaling pathways of their actions are currently in progress.

## Figures and Tables

**Figure 1 molecules-27-05908-f001:**
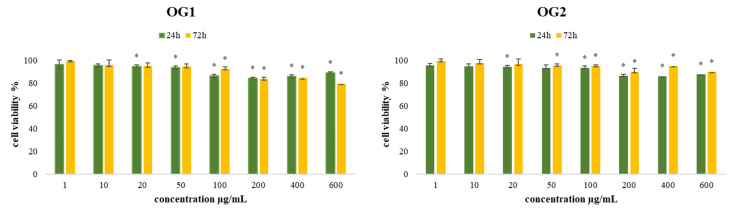
Effects of eight concentrations of OG1 and OG2 on MRC-5 cell viability after 24 h and 72 h of treatment. Results are presented as the mean of three independent experiments ± standard error; * *p* < 0.05 relative to control.

**Figure 2 molecules-27-05908-f002:**
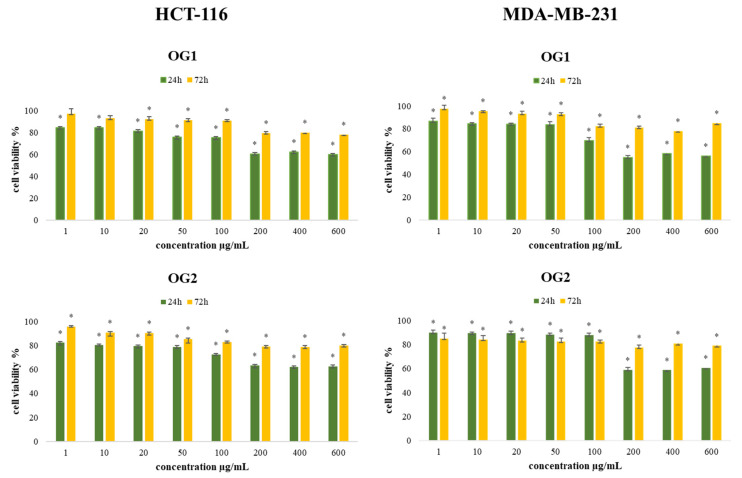
Effects of eight concentrations of essential oils (OG1 and OG2) on HCT-116 and MDA-MB-231 cells viability after 24 h and 72 h of treatment. Results are presented as the mean of three independent experiments ± standard error; * *p* < 0.05 relative to control.

**Figure 3 molecules-27-05908-f003:**
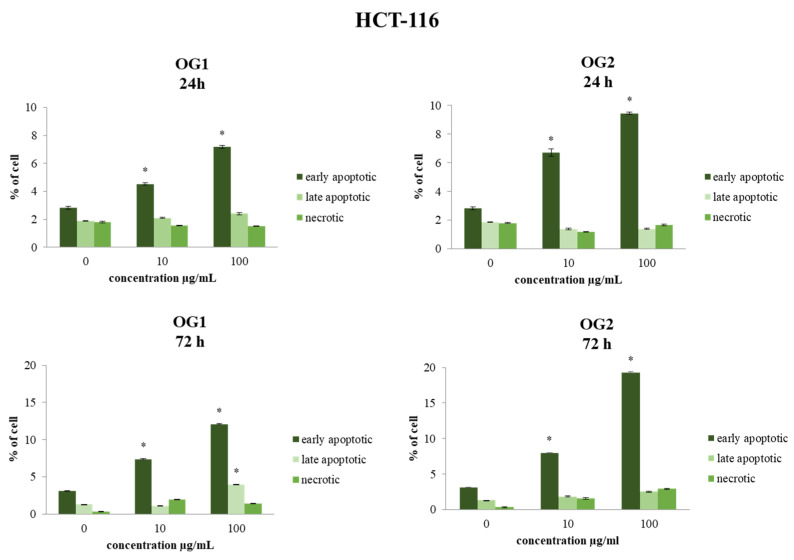
Flow cytometric analysis of Annexin V-FITC/7-AAD stained HCT-116 cells for 24 h and 72 h exposure with essential oils OG1 and OG2 at concentrations 10 µg/mL and 100 µg/mL. The percentages of early apoptotic (Annexin V+7-AAD−, lower right quadrant), late apoptotic (Annexin V+7-AAD+, upper right quadrant) and necrotic cells (Annexin V7-AAD+, upper left quadrant) in nontreated and treated cells are indicated on dot plots. Results are presented as the mean of three independent experiments ± standard error; * *p* < 0.05 relative to control.

**Figure 4 molecules-27-05908-f004:**
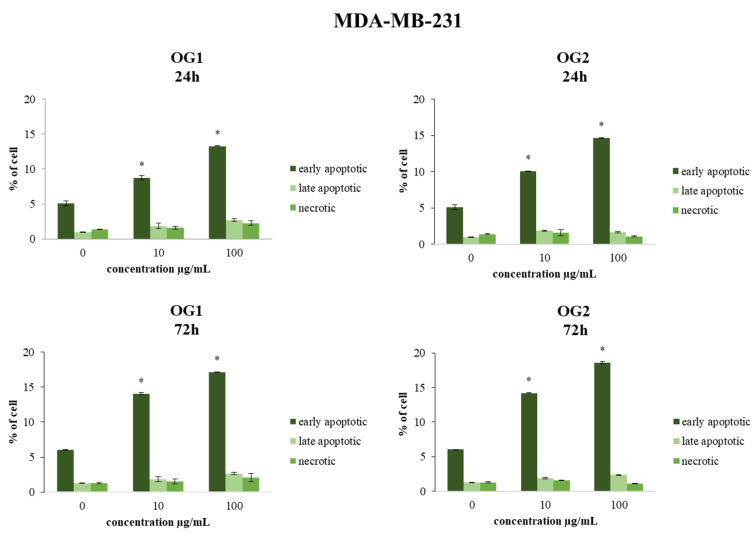
Flow cytometric analysis of Annexin V-FITC/7-AAD stained MDA-MB-231 cells for 24 h and 72 h exposure with essential oils OG1 and OG2 at concentrations 10 µg/mL and 100 µg/mL. The percentages of early apoptotic (Annexin V+7-AAD−, lower right quadrant), late apoptotic (Annexin V+7-AAD+, upper right quadrant) and necrotic cells (AnnexinV7-AAD+, upper left quadrant) in nontreated and treated cells are indicated on dot plots. Results are presented as the mean of three independent experiments ± standard error; * *p* < 0.05 relative to control.

**Figure 5 molecules-27-05908-f005:**
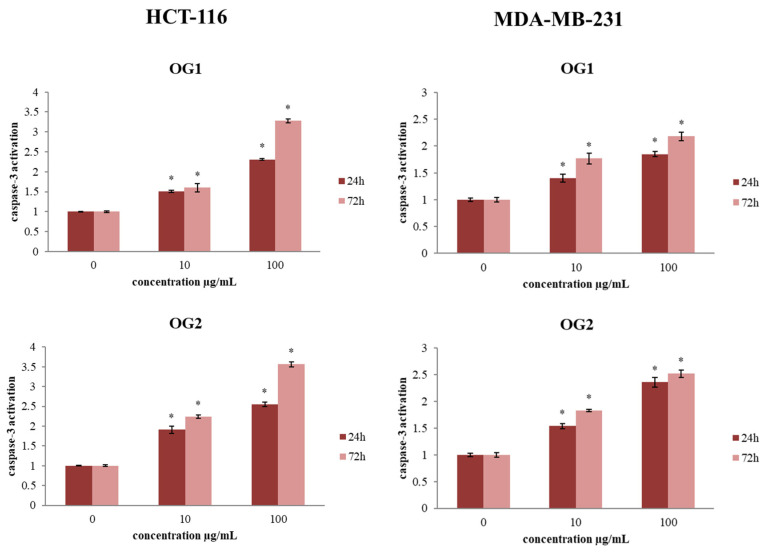
Antitumor activity of investigated essential oils against HCT-116 and MDA-MB-231 cells after 24 h and 72 h exposure with OG1 and OG2 oils at concentrations 10 µg/mL and 100 µg/mL compared to nontreated control cell. Caspase-3 activation was determined by using flow cytometry. Results are presented as the mean of three independent experiments ± standard error; * *p* < 0.05 relative to control.

**Figure 6 molecules-27-05908-f006:**
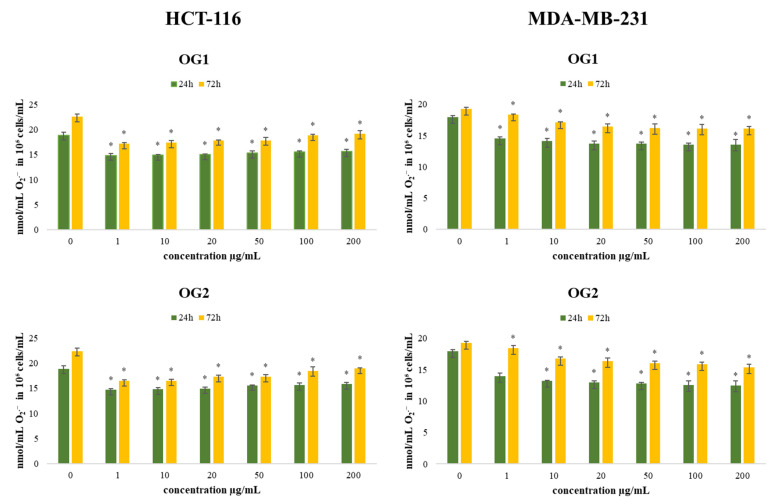
Effects of six concentrations of essential oils on the concentration of O_2_^•−^ in HCT-116 and MDA-MB-231 cells after 24 and 72 h of treatment. Results are presented as the mean of three independent experiments ± standard error; * *p* < 0.05 relative to control.

**Figure 7 molecules-27-05908-f007:**
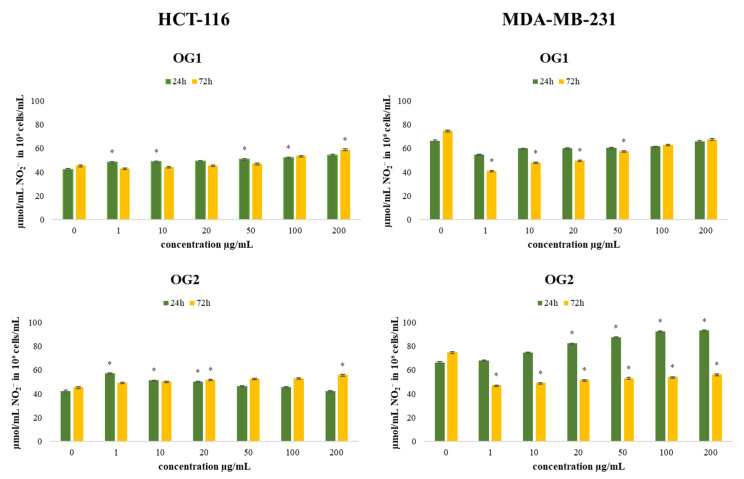
Effects of six concentrations of essential oils on the concentration of NO_2_^−^ in HCT-116 and MDA-MB-231 cells after 24 and 72 h of treatment. Results are presented as the mean of three independent experiments ± standard error; * *p* < 0.05 relative to control.

**Figure 8 molecules-27-05908-f008:**
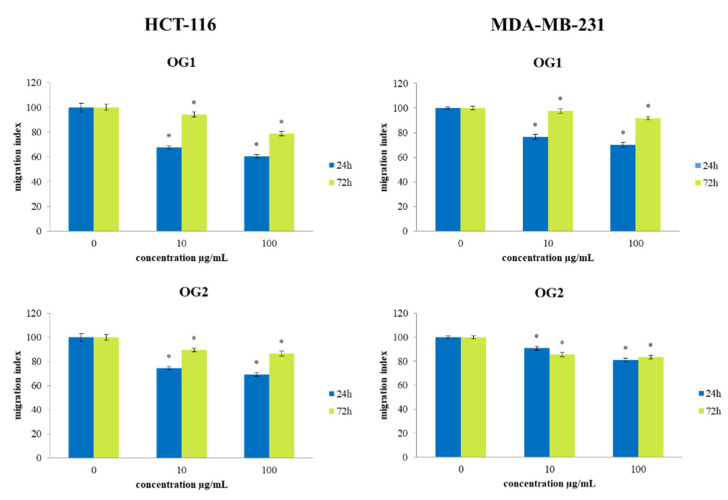
Effect of exposure to investigated essential oils OG1 and OG2 on migration index of HCT-116 and MDA-MB-231 cells. The cells were treated at concentrations of 10 µg/mL and 100 µg/mL during 24 h and 72 h exposure compared to nontreated control cell. Results are presented as the mean of three independent experiments ± standard error; * *p* < 0.05 relative to control.

**Table 1 molecules-27-05908-t001:** Chemical composition of essential oils from aerial parts of *O. grandiflora*.

No	RI ^a^	Compound ^b^	%	
			OG1	OG2
1	923	tricyclene	n.d. ^c^	tr ^d^
2	925	α-thujene	0.3	tr
3	934	α-pinene	4.5	1.4
4	949	camphene	0.2	0.6
5	951	thuja-2,4(10)-diene	tr	tr
6	973	sabinene	11.5	0.4
7	978	β-pinene	4.1	0.1
8	982	3-*p*-menthene	n.d.	0.1
9	984	myrcene	0.7	tr
10	1007	α-phellandrene	0.1	0.1
11	1016	α-terpinene	1.8	tr
12	1023	*p*-cymene	0.6	0.1
13	1030	β-phellandrene	0.8	0.8
14	1046	(*E*)-β-ocimene	0.1	n.d.
15	1057	γ-terpinene	3.6	0.1
16	1071	cis-linalool oxide	0.1	tr
17	1090	terpinolene	1.0	tr
18	1100	linalool	2.0	0.8
19	1174	terpinen-4-ol	9.6	0.5
20	1184	cryptone	n.d.	0.2
21	1188	α-terpineol	1.4	0.2
22	1194	myrtenal	0.1	n.d.
23	1208	(*E*)-dihydrocarvone	0.1	n.d.
24	1220	fenchyl acetate	0.1	n.d.
25	1274	phellandral	n.d.	0.1
26	1285	bornyl acetate	tr	2.5
27	1339	δ-elemene	n.d.	tr
28	1349	α-cubebene	0.2	0.1
29	1370	α-ylangene	0.1	tr
30	1377	α-copaene	0.8	0.4
31	1385	β-bourbonene	0.5	1.2
32	1390	β-cubebene	0.3	0.2
33	1392	β-elemene	1.1	22.7
34	1421	(*E*)-caryophyllene	5.5	6.8
35	1431	β-copaene	0.3	tr
36	1456	α-humulene	0.7	1.8
37	1458	(*E*)-β-farnesene	0.2	n.d.
38	1465	cis-muurola-4(14),5-diene	0.1	n.d.
39	1476	trans-cadina-1(6),4-diene	0.1	n.d.
40	1478	α-amorphene	n.d.	0.9
41	1480	γ-muurolene	1.0	0.3
42	1485	germacrene D	29.5	14.3
43	1489	β-selinene	0.2	2.3
44	1494	trans-muurola-4(14),5-diene	0.1	n.d.
45	1496	α-selinene	0.4	1.9
46	1498	bicyclogermacrene	1.3	1.5
47	1502	α-muurolene	0.6	0.2
48	1506	germacrene A	0.4	12.5
49	1509	δ-amorphene	0.2	n.d.
50	1517	γ-cadinene	0.6	0.4
51	1526	δ-cadinene	2.6	1.2
52	1535	trans-cadina-1,4-diene	0.1	n.d.
53	1540	α-cadinene	0.2	0.3
54	1557	germacrene B	0.4	1.1
55	1567	ledol	0.1	0.3
56	1578	spathulenol	0.9	2.6
57	1586	caryophyllene oxide	0.8	2.7
58	1607	b-oplopenone	0.2	0.5
59	1608	humulene epoxide II	n.d.	0.5
60	1614	1,10-di-epi-cubenol	n.d.	0.3
61	1620	junenol	0.3	0.1
62	1622	10-epi-g-eudesmol	n.d.	0.1
63	1629	1-epi-cubenol	0.1	0.1
64	1632	g-eudesmol	0.1	0.1
65	1638	isospathulenol	0.2	0.2
66	1644	τ-muurolol	1.3	1.4
67	1647	α-muurolol	0.2	0.2
68	1655	α-eudesmol	0.2	n.d.
69	1657	α-cadinol	1.5	0.2
70	1658	neo-intermedeol	n.d.	2.9
71	1661	7-epi-a-eudesmol	0.1	0.5
72	1667	14-hydroxy-(*Z*)-caryophyllene	n.d.	0.6
73	1669	(*Z*)-α-santalol	0.1	n.d.
74	1673	valeranone	n.d.	2.3
75	1688	germacra-4(15),5,10(14)-trien-1-a-ol	0.6	0.4
76	1692	eudesma-4(15),7-dien-1-b-ol	n.d.	0.2
77	1696	eudesm-7(11)-en-4-ol	0.6	0.8
78	1706	aromadendrene oxide II	n.d.	0.1
79	1710	(2*E*,6*Z*)-farnesal	n.d.	0.7
80	1714	(2*Z*,6*Z*)-farnesol	n.d.	0.5
81	1718	(2*E*,6*Z*)-farnesol	n.d.	1.3
82	1755	(2*E*,6*E*)-farnesol	n.d.	0.2
83	1766	14-oxi-a-muurolene	0.3	0.6
84	1773	14-hydroxy-a-muurolene	n.d.	0.1
85	1842	(2*E*,6*E*)-farnesyl acetate	0.2	0.1
86	1921	(5*E*,9*E*)-farnesyl acetone	0.2	n.d.
	**total**		**98.2**	**98.7**

^a^ Values of retention indices on HP-5MS column; ^b^ identified compounds; ^c^ n.d.—not identified; ^d^ tr—compounds identified in amounts less than 0.1%.

**Table 2 molecules-27-05908-t002:** Percentage composition and number of terpene hydrocarbons, oxygenated terpenes, and nonterpenoid compounds.

Class of Compound	OG1	OG2
	% (Number of Compounds)	
monoterpene hydrocarbons	29.3 (14)	3.7 (15)
oxygenated monoterpene	13.4 (8)	4.1 (6)
ketone	n.d. (0)	0.2 (1)
sesquiterpene hydrocarbons	47.5 (26)	70.1 (22)
oxygenated sesquiterpenes	8.0 (19)	20.6 (29)

n.d.—not identified.

**Table 3 molecules-27-05908-t003:** Class and percentage amounts of individual volatile compounds from essential oils of *O. grandiflora*.

Class of Compound	%	
	OG1	OG2
**monoterpenes**		
** *tricyclic monoterpenes* **		
*tricyclic tricyclene skeleton*	n.d.	tr
** *summ* **	**n.d.**	**tr**
** *bicyclic monoterpenes* **		
** *bicyclic cyclopropane thujane skeleton* **	11.8	0.4
*bicyclo [2.2.1]heptane camphane skeleton*	4.5	3.9
*bicyclic cyclobutane pinane skeleton*	4.4	0.7
*bicyclo [2.2.1]heptane fenchane skeleton*	0.1	n.d.
** *summ* **	**20.8**	**5.0**
** *monocyclic monoterpenes* **		
*monocyclic p-menthane skeleton*	19.0	2.0
*monocyclic 3,6-epoxy-2,6-dimethyloctane skeleton*	0.1	tr
** *summ* **	**19.1**	**2.0**
** *acyclic monoterpenes* **		
*acyclic 2,6-dimethyloctane skeleton*	2.8	0.8
** *summ* **	**2.8**	**0.8**
** *subtotal* **	**42.7**	**7.8**
**sesquiterpenes**		
** *tricyclic sesquiterpene* **		
*tricyclic 1,6-cycloguaiane skeleton*	0.5	0.3
*tricyclo [4.4.0.0(2,7)]dec-3-ene skeleton*	1.2	0.4
*cyclobuta(1,2:3,4)dicyclopentene skeleton*	0.5	1.2
*tricyclic 6,11-cycloguaiane skeleton*	1.2	3.1
*tricyclic 6,7-epoxy caryophyllane skeleton*	0.8	2.7
*tricyclic santalane skeleton*	0.1	n.d.
** *summ* **	**4.3**	**7.7**
** *bicyclic sesquiterpene* **		
*bicyclic caryophyllane skeleton*	5.5	7.4
*bicyclic cadinane skeleton*	9.2	6.7
*bicyclic eudesmane skeleton*	1.9	8.9
*bicyclogermacrane skeleton*	1.3	1.5
*bicyclic 6,7-epoxy-humulane skeleton*	n.d.	0.5
*bicyclic valerane skeleton*	n.d.	2.3
** *summ* **	**17.9**	**27.3**
** *monocyclic sesquiterpene* **		
*monocyclic elemane skeleton*	1.1	22.7
*monocyclic humulane skeleton*	0.7	1.8
*monocyclic germacrane skeleton*	30.9	28.3
** *summ* **	**32.7**	**52.8**
** *acyclic sesquiterpene* **		
*acyclic farnesane skeleton*	0.6	2.8
** *summ* **	**0.6**	**2.8**
** *tetracyclic sesquiterpene* **		
*tetracyclic 10,14-epoxy-6,11-cycloguaiane skeleton*	n.d.	0.1
** *summ* **	**n.d.**	**0.1**
** *subtotal* **	**55.5**	**90.7**
**nonterpenes**		
*ketone*	n.d.	0.2
** *subtotal* **	n.d.	0.2
**total**	**98.2**	**98.7**

n.d.—not identified.

## Data Availability

Data are contained within the article.

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
