# Peer review of "Chemical Composition, Antitumor Potential, and Impact on Redox Homeostasis of the Essential Oils of Orlaya grandiflora from Two Climate Localities"

_molecules, 2022, doi:10.3390/molecules27185908_

Round 1

Reviewer 1 Report (Previous Reviewer 2)

The authors have followed my recommendations and the article has a very good finished look for publication.

Author Response

Reviewer 1

The authors have followed my recommendations and the article has a very good finished look for publication.

We would like to thank the Reviewer for the time devoted for constructive and important comments to improve our paper.

Reviewer 2 Report (New Reviewer)

Despite the fact that the authors have done a lot of work, in my opinion the article has serious flaws.

- The main problem is the lack of proper control in all studies. There is an information that the untreated cells cultured only in DMEM served as control. If we consider that the stock solution was prepared at the concentration of 10 mg/mL,  then the highest concentration of DMSO in the cell culture was at 2% (for a OG concentration of 200 µg/ml, 6% !!! for 600 µg/ml). It is far too high concentration - for many cell lines 1 – 2% can be toxic. In addition, DMSO is a potent antioxidant, and the authors studied oxidoreductive processes. For these reasons, cells treated with DMSO alone at the appropriate concentration should be used as control cultures.

 - In my opinion, the authors draw too far-reaching conclusions from the results obtained. First of all, in the MTT test in the range of concentrations used, OGs  did not reach the IC50 concentration. This is not an optimistic result if we are expecting anticancer properties. Secondly, in most cases, after 72 hours, the cytotoxic effect is weaker (which the authors did not comment on).

Apoptosis/necrosis analysis shows little increase in the population of early-apoptotic cells. I understand that an increase of 5 - 10% was statistically significant, but is it really biologically important? To be honest, the observed effect is rather weak. Even after 72 hours of incubation, no late-apoptotic cells appeared

Additional remarks

- Authors need to decide whether they studied caspase 3 activity or expression

- There is no information about methods of statistical analysis

- The authors have interchangeably used the terms: proliferation and viability – these are two different phenomenon.

- What was the actual range of concentrations used in the study - concentrations of 400 and 600 µg/ml  appear and disappear in particular sections of the manuscript

- The choice of concentrations of 10 and 100 µg/ml is not entirely clear to me (average low and average high doses???)

- The description of the results does not always coincide with the corresponding graph (e,g. 341-44 In HCT-116 cells, all applied concentrations, at both time treatments, induced the increase in NO production compared to control, especially in the lowest concentration ……

Author Response

Reviewer 2

Respected Reviewer,

Thank you for your very useful comments and suggestions, they undoubtedly have improved the quality of the manuscript.

Coment of Reviewer:

Despite the fact that the authors have done a lot of work, in my opinion the article has serious flaws.
- The main problem is the lack of proper control in all studies. There is an information that the untreated cells cultured only in DMEM served as control. If we consider that the stock solution was prepared at the concentration of 10 mg/mL,  then the highest concentration of DMSO in the cell culture was at 2% (for a OG concentration of 200 µg/ml, 6% !!! for 600 µg/ml). It is far too high concentration - for many cell lines 1 – 2% can be toxic. In addition, DMSO is a potent antioxidant, and the authors studied oxidoreductive processes. For these reasons, cells treated with DMSO alone at the appropriate concentration should be used as control cultures.

Response to Reviewer: Thank you for your observation. We conduced this experiment correctly, but it was not written understandably. In the Part Matherials and Methods, section 3.5. Cell culture and treatment we clarified this. Please see lines 427-428.

The initial concentration of DMSO was 10%, so the final DMSO concentration in the highest examined dose of 600 µg/ml was 0.6%. In concentraion of 200 µg/ml, which was the higest dose used for all other parameters, the percent of DMSO was 0.2% or less. In numerous previous articles we tested the effects of these low DMSO concentration on the parameters of cell physiology, and no significant differences were observed, so we suggest that the presented control are reliable value for comparisons and our conclusions.

Coment of Reviewer:

 - In my opinion, the authors draw too far-reaching conclusions from the results obtained. First of all, in the MTT test in the range of concentrations used, OGs  did not reach the IC50 concentration. This is not an optimistic result if we are expecting anticancer properties. Secondly, in most cases, after 72 hours, the cytotoxic effect is weaker (which the authors did not comment on).

Response to Reviewer: Not all natural product could rech IC50 concentration. The antitumor activity was not only measured by MTT test, but also with caspase 3 acitivty, apoptosis ratio and migration capacity. In all prameters the significant drops were recorded indicating these essential oils as promising antitumor agents for some therapeuic cotreatments. After 72h the drop of MTT values are still considerable, eventhough the cells may acquire some resistance on OGs effects by unknown mecanisms.

Coment of Reviewer:

Apoptosis/necrosis analysis shows little increase in the population of early-apoptotic cells. I understand that an increase of 5 - 10% was statistically significant, but is it really biologically important? To be honest, the observed effect is rather weak. Even after 72 hours of incubation, no late-apoptotic cells appeared.

Response to Reviewer: The proapototic effects are around 3 fold higher compred to control in 24h, and in 72 h the effects are even stronger (almost 6,20 fold higher in OG2), and the caspase activity is more then 3.56 fold higher. Based on these data we think that we can indicate that the detected biological effects are not unimportant, especialy considering that in literature there are no previous results concernig OG1 and OG2 biological effects.

Coment of Reviewer:

Additional remarks
- Authors need to decide whether they studied caspase 3 activity or expression

Response to Reviewer: Thank you very much for your valuable suggestion. We changed the terms accordigly to your suggestion. Now through the manuscript we used activation instead of expression. Please see lines  36, 264 and 425.

Coment of Reviewer:

- There is no information about methods of statistical analysis

Response to Reviewer: Thank you very much for your valuable suggestion. We added the paragraph considering the statistical analysis in the manuscript according to your suggestion. Please see section 3.12. Statistical analyses (Last paragraph in the Part Materials and Methods).

Statistical analyses

All data were evaluated using IBM-SPSS 23 software for Windows (SPSS Inc., Chicago, IL, USA). The data were presented as a mean ± standard error (S.E.M). The statistical significance was determined using a Paired Sample - T test. The level of statistical significance was set at *p < 0.05.

Coment of Reviewer:

- The authors have interchangeably used the terms: proliferation and viability – these are two different phenomenon.

Response to Reviewer: Thank you very much for your valuable suggestion. We have rewritten the terms accordigly in the manuscript. Now through the manuscript the term viability was used. Please see lines 32, 192 and 277.  

Coment of Reviewer:

- What was the actual range of concentrations used in the study - concentrations of 400 and 600 µg/ml  appear and disappear in particular sections of the manuscript.

Response to Reviewer: Thank you very much for your valuable suggestion. The concentrations of 400 and 600 µg/mL were only used in MTT test, and since no significant difference was recorded comparing with concentrations equal and below 200 µg/mL, in all other parameters we have used  200 µg/mL as maximal concentration.

Coment of Reviewer:

- The choice of concentrations of 10 and 100 µg/ml is not entirely clear to me (average low and average high doses???)

Response to Reviewer: Thank you very much for your valuable suggestion. The concentrations of 400 and 600 µg/mL were only used in MTT test, and since no significant difference was recorded in all other parameters maximal concentration of 200 µg/mL was used. Considering that, in a group of low doses (1 µg/mL, 10 µg/mL, and 20 µg/mL) we used concetration of 10 µg/mL as representative. Also, for a group of high doses (50 µg/mL, 100 µg/mL, and 200 µg/mL) we used concetration of 100 µg/mL as representative one.

Coment of Reviewer:

- The description of the results does not always coincide with the corresponding graph (e,g. 341-44 In HCT-116 cells, all applied concentrations, at both time treatments, induced the increase in NO production compared to control, especially in the lowest concentration ……

Response to Reviewer: Thank you very much for your valuable suggestion. Please see lines 330-333. The results correspond with the stated graph, and we added the type of the essential oils (OG2) in order to make the statement clear.

Round 2

Reviewer 2 Report (New Reviewer)

The authors have replied to most of my concerns. I leave judgment on the significance of these results to the readers.

This manuscript is a resubmission of an earlier submission. The following is a list of the peer review reports and author responses from that submission.

Round 1

Reviewer 1 Report

The authors presented the chemical compositions of 2 aromatic plants in Serbia, and determined their cytotoxicity, and redox activities. The manuscript is fine and well organized, just one comment plz change the style of the letters para, ortho into italic.

Reviewer 2 Report

Dear authors,

Natural products and plant extracts are extremely interesting in the field of their antitumor activity. You study the concentration dependence on cell viability and its decrease by about 40%. Why did you stop at this concentration?

These studies are not enough grounds to consider that the extracts have an antitumor effect. Furthermore, you provide data, albeit not convincing enough, at an effective concentration of 200 micrograms per milliliter. Why didn't you continue in the higher concentration range? Then the results would be much more convincing. Based on such results, you cannot present IC50 values. In a possible follow-up study, you should treat the cells with the established IC50 or EC50 and then already determine the superoxide or nitro radical content.

As such, the obtained results are not sufficient grounds to consider that the extracts have an antitumor effect.

The results would be more conclusive if some of the methods of apoptosis on the cell line, or the caspases responsible for apoptosis, were examined at the concentration where you get the most severe inhibition of viability.

It is considered that superoxide is oncogenic, it is good that you measured the level of generated superoxide. That the decrease is about 10% at 72 hours IS not reliable evidence, because 10% could only be the error of the spectrophotometric method. The method you used does not show whether the level of intracellular superoxide is increased, which indeed has been shown to have an oncogenic effect

Your study needs additional evidence to accept the antitumor effect of the two extracts you studied.